# Titanium Dioxide, but Not Zinc Oxide, Nanoparticles Cause Severe Transcriptomic Alterations in T98G Human Glioblastoma Cells

**DOI:** 10.3390/ijms22042084

**Published:** 2021-02-19

**Authors:** Encarnación Fuster, Héctor Candela, Jorge Estévez, Eugenio Vilanova, Miguel A. Sogorb

**Affiliations:** Instituto de Bioingeniería, Universidad Miguel Hernández de Elche, Avenida de la Universidad s/n, 03202 Elche, Spain; e.fuster@umh.es (E.F.); hcandela@umh.es (H.C.); jorge.estevez@umh.es (J.E.); evilanova@umh.es (E.V.)

**Keywords:** titanium dioxide nanoparticles, zinc oxide nanoparticles, T98G human glioblastoma, nanotoxicology, omics, neurotoxicity, neuroinflammation, cell toxicity, toxicology, human health

## Abstract

Titanium dioxide and zinc oxide are two of the most widely used nanomaterials. We assessed the effects of noncytotoxic doses of both nanomaterials on T98G human glioblastoma cells by omic approaches. Surprisingly, no effects on the transcriptome of T98G cells was detected after exposure to 5 µg/mL of zinc oxide nanoparticles during 72 h. Conversely, the transcriptome of the cells exposed to 20 µg/mL of titanium dioxide nanoparticles during 72 h revealed alterations in lots of biological processes and molecular pathways. Alterations to the transcriptome suggests that exposure to titanium dioxide nanoparticles might, potentially, compromise the integrity of the blood brain barrier integrity and cause neuroinflammation. The latter issue was further confirmed phenotypically with a proteomic analysis and by recording the release of interleukin 8. Titanium dioxide also caused autophagy, which was demonstrated through the increase in the expression of the autophagy-related 3 and microtubule associated protein 1 light chain 3 alpha genes. The proteomic analysis revealed that titanium dioxide nanoparticles might have anticancerigen properties by downregulating genes involved in the detoxication of anthracyclines. A risk assessment resulting from titanium dioxide exposure, focusing on the central nervous system as a potential target of toxicity, is necessary.

## 1. Introduction

All particles with at least one dimension below 100 nm are typically considered nanoparticles (NPs) [1]. Nanomaterials (NMs) usually display quite different physical properties to their equivalents on the non-nanometric scale. Of all the technologies developed in the 21st century, nanotechnology offers the highest potential to provide social welfare and progress and can influence many fields, such as biomedicine, cosmetics, the food industry and the environment [2].

The current European Union (EU) market contains 340 NMs [2]. The largest online inventory (developed in 2015) lists 1814 nanotechnology consumer products from 622 companies [3]. These figures allow us to understand that human beings are already exposed to NPs of anthropogenic origin and that such exposure will very likely increase in the near future. Therefore, a careful safety assessment of biotechnological-based nanotechnology applications for the environment and human health is desirable.

Titanium dioxide (TiO_2_) and zinc oxide (ZnO) are two of the most widely used NMs. Indeed, the tonnage band for TiO_2_ NMs is above 1,000,000 tons per year. In Europe, more than 200 entries in the Registration, Evaluation, Authorisation and Restriction of Chemicals Regulation (REACH) [4] for TiO_2_ are noted. Moreover, according to the REACH registry, between 100,000 and 1,000,000 of tons per year of ZnO NMs per year are produced for close to 200 different active registrations [4].

Both TiO_2_ and ZnO NMs have similar applications. Some involve little or negligible human exposure when used as pigments, coverings or in electronics. However, other applications cause considerable concern from the human exposure point of view. For example, TiO_2_ nanoparticles (TiO_2_-NPs) and ZnO nanoparticles (ZnO-NPs) are used in 43 and 25 different categories of cosmetics, respectively [5]. Biomedical applications in imaging diagnostics, such as drug carriers or dental composite resins, together with applications in food industry packaging or as taste carriers or sensors [2,6,7], are matters of serious concern about human exposure. Moreover, the Organization for Economic Cooperation and Development (OECD) has included TiO_2_-NPs and ZnO-NPs among those NMs whose safety should be addressed as a priority [8].

Several authors highlighted that engineered NMs can cause neurotoxicity [9]. Both ZnO-NPs and TiO_2_-NPs are able to cross the blood–brain barrier (BBB) [10,11,12] and are, therefore, potentially able to induce neurotoxicity. Indeed, morphological changes of neurons, increases in dopamine and norepinephrine secretions and oxidative stress and reductions in cell proliferation in the hippocampus, together with impaired learning and memory, have been reported when offspring have been exposed to TiO_2_-NPs [13]. The impairment of synaptic responses and disrupted spatial memory have also been reported as consequences of ZnO-NPs [13]. However, no reports about neurotoxicity mediated through glia alterations can be found. Glia is the most abundant cell line in the central nervous system (CNS) and is involved in its homeostasis and other neurological processes, such as regulation of the BBB, neural plasticity and myelination [14]. Thus, the effects of NPs on glia cells should be addressed, because alterations to the glia performance might potentially lead to neurotoxicity in the CNS; for example, by increasing the bioavailability of other neurotoxicants to neurons since glia cells form part of the BBB.

We previously studied the effects of silver NPs on T98G human glioblastoma cells and found that these NPs are able to alter the secretion of interleukins and the fibroblast growth factor (FGF) pathway (in the latter, this is probably achieved by mitogen-activated protein kinase (MAPK) cascade dysregulation) [15]. In this work, we studied the effects of TiO_2_-NPs and ZnO-NPs by microscopic, Enzyme-Linked ImmunoSorbent Assay (ELISA), transcriptomic and proteomic approaches and observed notable differences between the two NMs, as TiO_2_-NPs seemed much more deleterious for T98G human glioblastoma cells than ZnO-NPs.

## 2. Results

### 2.1. Physicochemical Properties of Nanoparticles

The distribution size of the suspensions of both NPs in type I water was determined by dynamic light scattering (DLS) (Figure 1A,B). The most represented sizes were 707.9 ± 173.3 and 786.9 ± 176.7 nm for ZnO-NPs and TiO_2_-NPs (Table 1), respectively. Wide deviations appeared, which showed that colloidal stability was lacking (Figure 1A,B). Indeed, both suspensions displayed turbidity and NP decantation after a few minutes of rest. The samples of these same suspensions showed Z-potentials of +17.0 ± 0.6 mV (for ZnO-NPs) and +22.8 ± 0.8 mV (for TiO_2_-NPs) (Table 1). Appendix A provides the results obtained in the three determinations of the Z-potentials per NM.

Figure 1C,D shows the distribution sizes of the NPs determined by transmission electron microscopy (TEM). The sizes with the highest percentage of NPs were 23.15 ± 8.65 and 18.18 ± 5.25 nm for Zn-NPs and TiO_2_-NPs, respectively (Table 1). The TEM analysis of the Zn-NPs showed that these NPs came in different shapes (spheres, ovals, sticks and other nongeometric shapes) with high polydispersity (Figure 1E).

#### Stability of NPs in Water and Cell Culture Medium

For ZnO-NPs, the main particle size in water at time 0 h was 712 nm (Figure 2), while the size is notably reduced when this NM came into contact with cell culture media, because two main populations with the largest proportion of nanoparticles were found at 6.5 and 12 nm (Figure 2). After 24 h of incubation in water, the main nanoparticle size became slightly smaller and two well-defined populations of nanoparticles appeared with the most represented sizes of 342 and 615 nm (Figure 2). This reduction in NP sizes in water was time-dependent, because three populations of NPs were identified with maximum proportions at 1.1 nm, 7.5 nm and 342 nm after 72 h of incubation (Figure 2). This size reduction phenomenon with time became much more notable with the ZnO-NPs suspended in the cell culture medium, as the more represented sizes after 24 h and 72 h of incubation were 5.6–11.7 and 4.2–10.1 nm, respectively (Figure 2). The decantation of NPs observed in the cell culture media when the suspension was left to rest was noteworthy, although this phenomenon was slower than when the particles were dispersed in water.

For TiO_2_-NPs, the particle size in water at the beginning of the experiment was 615 nm, while several smaller-sized populations appeared when this same NM was dispersed in cell culture media, which suggests that NPs were dispersed (Figure 3). In this case, the effect was more evident after 24 h of incubation, when the size with the most NPs was 6.5 nm in the cell culture media and 825 nm in water (Figure 3). The aggregation of TiO_2_-NPs in water was even more remarkable after 72 h of incubation (mean size of 955 nm) (Figure 3). The mean size of the NPs after 72 h of incubation in the cell culture medium increased until it reached a maximum of 12 nm (Figure 3). As with ZnO-NPs, a certain decantation of TiO_2_-NPs was observed, and this process was faster in water than in the cell culture media.

### 2.2. Uptake of NPs in T98G Glioblastoma Cells

#### 2.2.1. Microscopy

Cells were exposed for 72 h to 20 µg/mL of TiO_2_-NPs or 5 µg/mL of ZnO-NPs, as described in Section 4.4, and were then fixed for the electronic microscopy assessment, as described in Section 4.7. The results with SEM are presented in Appendix A. No significant differences were detected between the control cultures and the cultures treated with ZnO-NPs and TiO_2_-NPs, which suggests that NPs did not come into contact with the cells on the exterior plasmatic membrane and that exposure to both NPs caused no apparent alterations to the surfaces of plasmatic membranes.

The TEM analysis results are presented in Figure 4. No significant differences were found between the control and ZnO-NP-exposed cells. However, TiO_2_-NPs were observed inside T98G cells (Figure 4). TiO_2_-NPs were grouped and formed clusters inside the cytoplasm, and signs of apparent autophagy were reported all around these clusters of TiO_2_-NPs (Figure 4). These autodigestion signs were not found in the control or ZnO-NP-exposed cultures.

#### 2.2.2. Flow Cytometry

T98G human glioblastoma cells were firstly exposed to either 20 µg/mL of TiO_2_-NPs or 5 µg/mL of ZnO-NPs, as described in Section 4.4. Next, the viability and light scattering were analyzed by flow cytometry, as described in Section 4.6. The side scatter (SSC) intensity is proportional to the cell granularity, which should be increased if NPs are incorporated into cells or remained bound to cytoplasmic membranes. No significant differences were found between the SSC records obtained in the control cultures and the cultures exposed to ZnO-NPs for 24 h and 72 h (Table 2). However, the SSC was notably increased for the T98G cells exposed to 20 µg/mL of TiO_2_-NPs (Table 2). Moreover, this effect was time-dependent, and the effect of NPs on the SSC increased 1.7-fold between 24 h and 72 h (Table 2). The cell viability of cultures was above 90% (Table 2) in all cases and was, therefore, similar to the viability determined by 3-(4,5-dimethylthiazol-2-yl)-2,5-diphenyltetrazoliumbromide (MTT) (Table 3). Hence, these experiments suggest that TiO_2_-NPs, but not ZnO-NPs, remain bound or inside T98G cells after exposure.

### 2.3. Effects of NPs on T98G Cell Viability

The cytotoxicity of NPs was assessed by MTT tests in order to determine the maximum dose that would cause viability to lower no more than 10% to be used in further experiments as massive parallel RNA sequencing (RNAseq) and others. This dose was set at 5 µg/mL for ZnO-NPs, because this dose was able to reduce the cell viability to 92% ± 1% (*n* = 4) after 72 h of exposure, as described in Section 4.4 of the Material and Methods (Table 3). A higher dose (10 µg/mL) of these NPs reduced the cell viability by 13% (Table 3), which was considered excessive cytotoxicity. The exposure of T98G human glioblastoma cells to up to 20 µg/mL of TiO_2_-NPs under the conditions described in Section 4.4 of the Material and Methods reduced the cell viability by between 7% and 10% (Table 3), while 30 µg/mL of TiO_2_-NPs reduced the viability by 13% (Table 3). Thus, 20 µg/mL of TiO_2_-NPs was set to be used in further experiments. 

### 2.4. Massive Parallel Sequencing of RNA of the Cells Exposed to NPs

T98G human glioblastoma cells were exposed to either 20 µg/mL of TiO_2_-NPs or 5 µg/mL of ZnO-NPs for 72 h, as described in Section 4.4. Each experimental condition (non-exposed and exposed cells) was assayed in triplicate. The RNA of the control and exposed cells was further isolated, as described in Section 4.8. The quality of the RNA samples was analyzed by an Agilent Bioanalyzer 2100 to obtain an RNA integrity number (RIN) for the nine samples of 9.5 ± 0.4 (mean ± SD) (range: 8.7–9.8). These RNA samples were sent to Macrogen Inc. (South Korea) and were analyzed again for RNA quality and gave RIN values of 7.9 ± 0.5 (mean ± SD) (range: 7.0–8.8). Despite the reduction in RNA quality, all the samples reached a RIN that equaled or exceeded 7.0, the typical value used as the threshold for considering a sample to be apt for RNAseq. Indeed, the three well-defined peaks corresponding to ribosomal RNA molecules 5S, 18S and 28S were clearly seen in all the samples (for an example of the electropherograms used to estimate RIN, see Appendix A). The RNAseq run on the Illumina next-generation sequencing (NGS) platform was performed by MACROGEN Inc. The results were analyzed using bioinformatic tools, as described in Section 4.10.

Appendix A shows the total number of reads obtained for each sample, the amount of sequences obtained, their guanine-cytosine content and the proportion of reads with average Phred scores that equaled or exceeded 20 (Q20) or 30 (Q30). As expected, the guanine-cytosine content was similar in all the studied samples, and the percentages of reads with Q20 and Q30 values were always higher than 98% and 95%, respectively. More than 98% of the reads from each sample aligned to the reference genome, with most read pairs aligning concordantly (data not shown). The frequency of duplicate reads in each sample ranged between 26.88% for replicate 3 of the control cells and 41.51% for replicate 3 of the ZnO-NPs-treated cells.

#### 2.4.1. Differentially Expressed Genes

The statistical analysis showed that no genes were differentially expressed in the T98G cells exposed to ZnO-NPs. Conversely, the bioinformatic analysis set a 5% false discovery rate (FDR) threshold to allow the identification of the 1025 genes expressed at significantly different levels in the control and TiO_2_-NP-exposed cells (Appendix A). Five hundred and seventeen genes were upregulated, while the remaining 508 genes were downregulated (Appendix A). The gene with the highest downregulation exhibited a fold change (FC) of 0.14 (log_2_ FC = −2.79) (Appendix A). This gene was ENSG00000164188, which encodes a nuclear factor (RAN-binding protein 3-like) involved in both the termination of bone morphogenetic protein signaling and the regulation of mesenchymal stem cell differentiation [16]. Moreover, the exposure of T98G cells to TiO_2_-NPs lowered the expression of gene ENSG00000275479, which went below the limit of detection (Appendix A), and this gene (for which no protein transcript was found [17]) was still detected in the control cells. On the contrary, the gene with the highest upregulation (log_2_ FC = 4.06) was ENSG00000135253 (Appendix A). This gene encodes the kielin/chordin-like protein, which enhances bone morphogenetic protein signaling in a paracrine manner and inhibits both the activin-A and transforming growth factor β1 (TGFB1)-mediated signaling pathways [18].

#### 2.4.2. Annotation of the Differentially Expressed Genes

The set of the differentially expressed genes contained 1025 genes. Despite statistical significance, the level of dysregulation was relatively moderate. Therefore, in order to better understand the functions altered by exposure to TiO2-NPs, we decided to use Gene Ontology (GO) Consortium tools to analyze only those genes with an FC that either equaled or exceeded 2 (log_2_ FC ≥ 1) or equaled or was below 0.5 (log_2_ FC ≤ −1). These thresholds shortened the list to be uploaded to the PANTHER Classification System tool to 208 genes, of which 126 were downregulated (green cells in Appendix A) and 82 were upregulated (orange cells in Appendix A). We used as a control the 11,869 genes expressed in the control cells (Appendix A), where 11,260 of them were assigned at least one GO term. Besides, 195 genes of the differently expressed genes were also assigned to at least one GO term.

An enrichment analysis based on Fisher’s exact test with an FDR threshold of 0.05 for controlling false positives was run to identify the overrepresented terms among the differentially expressed genes. The results of the “*slim biological process*” and the “*slim molecular function*” sub-ontologies are presented in Table 4 and Table 5, respectively.

Table 4 and Table 5 show how the exposure of T98G human glioblastoma cells to TiO_2_-NP could alter the performance of these cells as regards their immunological and the BBB roles. The altered biological processes and biological functions addressing the immune system were: “*granulocyte chemotaxis*” (GO:0071621) (the biological process with the highest enrichment factor), “*response to lipopolysaccharide*” (GO:0032496) (the biological process with the second-highest enrichment factor), “*response to cytokine*” (GO:0034097) (Table 4) and “*cytokine activity*” (GO:0005125) (the molecular function with the second-highest enrichment factor) (Table 5). The alterations to the biological process “*cell adhesion*” (GO:0007155) (enrichment factor of 5) (Table 4) and in the molecular function of “*integrin binding*” (GO:0005178) (enrichment factor of 16) (Table 5) might address the role of glia cells in the BBB.

#### 2.4.3. Altered Molecular Pathways

The altered molecular pathways were also analyzed by PANTHER, which reported 40 altered pathways. Table 6 shows only those pathways with at least two dysregulated genes on the list of the 208 uploaded for the GO term analysis. Table 6 shows several pathways related to the immunological functions associated with glia cells, such as the “*interleukin signaling pathway*”, “*inflammation mediated by chemokine and cytokine signaling pathway*”, “*B-cell activation*” and “*T-cell activation*”, together with other pathways associated potentially with maintaining the BBB integrity, such as the “cadherin signaling pathway” and “integrin signaling pathway”. 

The proto-oncogene c-Fos was involved in the most pathways (up to seven) (Table 6). This gene encodes a nuclear phosphoprotein with an important role in signal transduction, cell proliferation and differentiation [19]. The gene with the second-largest number of involvements in the altered pathways (up to five) was the nuclear factor of activated T-cell 4, which codes for a Ca^2+^-regulated transcription factor that is involved in several processes, including the development and function of the immune, cardiovascular, musculoskeletal and nervous systems [20]. Five other genes were involved in three different pathways; 8 other genes in 2 pathways and, finally, 24 genes were involved in a single pathway alteration (Table 6).

### 2.5. Proteomic Alterations in the T98G Cells Exposed to TiO_2_-NPs

T98G human glioblastoma cells were exposed to 40 µg/mL of TiO_2_-NPs for 72 h, as described in Section 4.4. Afterwards, cells were washed to remove NPs and were fractionated, as described in Section 4.12. Protein preparations were transported in an ice bath to the Scientific and Technical Research Area of the University of Murcia (Spain) (ACTI) where a proteomic analysis was performed, as described in Section 4.12. Each experimental condition was assayed with three biological replicates. The raw proteomic analysis data are shown in Appendix A. The number of proteins identified in the cytoplasmic and membrane fractions of the control cells was, respectively, 305 and 304, while the number of peptides was 20,488 in the cytoplasmic fraction and 11,868 in the membrane fraction (Appendix A). A similar number of peptides (11,868) and proteins (310) to those identified in the cytoplasmic fraction of the control cells was found in the cytoplasmic fraction of the TiO_2_-NP-treated cells (Appendix A). However, the number of proteins (271) and peptides (10,488) identified in the membrane fractions of the TiO_2_-NP-treated cells was slightly smaller than those reported in the membrane fraction of the control cells (Appendix A).

Appendix A shows the lists of proteins whose expression were statistically altered after exposure to TiO_2_-NPs. Nineteen proteins were dysregulated in the cytoplasmic fraction. Of them, six were upregulated between 1.35- and 2.29-fold, two were detected in the treated cells but not in the control cells, one was detected in the control cells but not in the treated cells and 10 were downregulated between 0.42- and 0.70-fold (Appendix A). The number of proteins identified in the membrane fraction (37) was higher than the number of proteins identified in the cytoplasmic fraction. All the dysregulated proteins belonging to the membrane fraction were downregulated between 0.05- and 0.5-fold, except for the three proteins detected exclusively in the control cells and the two detected exclusively in the treated cells (Appendix A).

Table 7 shows the result for the biological process sub-ontology of the enrichment analysis (Fisher’s exact test with an FDR threshold of 0.05) performed by GO tools. Three different biological processes were found with an enrichment factor higher than 100 (“*doxorubicin metabolic process*” (GO:0044598), “*daunorubicin metabolic process*” (GO:0044597) and “*interleukin 12-mediated signaling pathway*” (GO:0035722). “*Cell–cell recognition*” (GO:0009988) also was found with a notable enrichment factor (70) (Table 6). Two other biological processes were found with enrichment factors between 10 and 30 and three with enrichment factors below 10 (Table 6).

### 2.6. Validation of Massive Parallel Sequencing by PCR

Three different genes (two with an altered expression and one with an unaltered expression) were selected to validate the RNAseq results by real-time quantitative polymerase chain reactions (RT-PCR). The genes with a statistically altered expression level were JUN B and IL8. These genes were involved in three of the altered molecular pathways displayed in Table 6. The gene with no altered expression after exposure to TiO_2_-NPs was interleukin 6 (IL6), a homologous of IL8 whose expression was altered in these cells by exposure to silver NPs [15].

#### 2.6.1. Selecting Housekeeping Genes

We tested the effect of TiO_2_-NPs on the expression of 24 different human genes typically used as housekeeping genes (see Section 4.11). The results are shown in Appendix A. The TiO_2_-NPs had no statistically significant effect on the expression of 21 of the 24 assessed genes, while the number of thermal cycles required to reach the fluorescence threshold in the PCR increased statistically for 18S rRNA and dropped for UBS and GADD45A (Appendix A). Housekeeping genes GAPDH (glyceraldehyde 3-phosphate dehydrogenase) and PGK1 (phosphoglycerate kinase 1) were selected, as their expression levels were not altered by either the TiO_2_-NPs or the silver NPs [15], and they have often been used for this purpose in the scientific literature.

#### 2.6.2. Expressions of the Selected Genes

The expression levels of the above-cited genes were assayed with two different housekeeping genes in two independent experiments. T98G human glioblastoma cells were exposed to TiO_2_-NPs under the same conditions as in the RNAseq experiments (20 µg/mL for 72 h). RNA was then isolated, as described in Section 4.8, and the expression level of each gene was tested by RT-PCR, as described in Section 4.11. Three different plates per experimental condition were used in each independent experiment.

The TiO_2_-NPs brought a statistically significant increase in the IL8 expression of around 2.6-2.8-fold in the samples assayed by both RNAseq and RT-PCR (with both housekeeping genes) (Table 8), while no alterations in the IL6 expression were detected by RT-PCR. The same occurred for RNAseq (Table 8). The TiO_2_-NPs also caused a statistically significant reduction of around 80–85% in the expression level of the JUN B gene (Table 8) when determined by RT-PCR and of around 58% when determined by RNAseq (Table 8). Therefore, the results of the two independent RT-PCR experiments were reproducible, and no differences were detected when the effects were assessed using GAPDH or PGK1 as the housekeeping genes (Table 8). Moreover, the effects of TiO_2_-NPs on the three tested genes were well-matched in the three independent experiments by two different measurement methodologies (RNAseq and RT-PCR) (Table 8).

#### 2.6.3. Effect of the Concentration of TiO_2_-NPs on Gene Expressions

T98G human glioblastoma cells were exposed to three different concentrations (20, 2 and 0.2 µg/mL) of TiO_2_-NPs for 72 h, as described in Section 4.4. When the exposure ended, RNA was isolated, as described in Section 4.8, and the expression of each gene was determined by RT-PCR using GAPDH as the housekeeping gene, as described in Section 4.11. Each experimental condition was assayed in three different biological replicates, and the results are shown in Figure 5. Only the highest concentration (20 µg/mL) had statistically significant effects on the expressions of IL8 (an increase of around three-fold) and JUN B (a reduction of approx. 58%) (Figure 5). As expected, and in line with the results shown in Table 8, no effect was found at any concentration for IL6 expression (Figure 5). The magnitude of the effects reported on the expressions of IL8 and JUN B after exposure to 20 µg/mL of TiO_2_-NPs was fully comparable to the effects reported in Table 8.

### 2.7. Autophagy Induced by TiO_2_-NPs

The TEM analysis suggested that the exposure of T98G cells to TiO_2_-NPs can induce autophagy (Figure 4). In order to test this hypothesis, we exposed cells for different times to 20 µg/mL of TiO_2_-NPs. Then, we isolated the RNA as described in Section 4.8 and assayed the expression of the genes BECLIN 1, microtubule-associated protein 1 light chain 3 alpha (LC3A) and autophagy-related 3 (ATG3) by RT-PCR, as described in Section 4.11. The expression of ATG3 and LC3A was increased between two- and three3-fold (*p* < 0.01) (Figure 6). These differences were not noted after 24 h of exposure (Figure 6). No significant variations in the expression of BECLIN1 were noted at any time.

### 2.8. Effect of NPs on the Secretion of IL6 and IL8

The expression of the gene encoding IL8, but not the gene encoding IL6, significantly increased after exposure to TiO_2_-NPs (Table 8 and Appendix A). We assayed the effect of exposing ZnO-NPs and TiO_2_-NPs on the secretion of IL8 by ELISA procedures, as described in Section 4.13. The excretion of IL8 after exposure to TiO_2_-NPs statistically increased by 1.4- to 1.5-fold after 24 h of exposure (Table 9). This increase in IL8 slightly reduced to 1.3- to 1.4-fold after 72 h of exposure, although the excretion of IL8 was still significantly higher than the excretion of the control cells (Table 9). The excretion of IL8 of the cells exposed to the positive control lipopolysaccharide from *Escherichia coli* O55:B5 (positive control) was always statistically higher than the excretion of IL8 in the control cells (Table 9). No consistent changes were detected in the excretion of IL8 after exposure to ZnO-NPs. Indeed, one of the two experiments lasting 24 h showed a 13% reduction in the excretion, which was not confirmed in the second experiment and at the same time. Moreover, two experiments showed no effects after 72 h of exposure, and a third experiment reported a slight increase (Table 9). In all cases, the effects reported after exposure to ZnO-NPs were lower in magnitude than the increases reported after TiO_2_-NP exposure (Table 9).

## 3. Discussion

The new paradigm for toxicity testing proposed by the US National Academy of Sciences [21] is based on the identification of biologically relevant molecular perturbations caused (preferably) in human cells as a first response to exposure. With this approach, we observed quite different effects of TiO_2_-NPs and ZnO-NPs on T98G human glioblastoma cells. No significant alterations to the transcriptome of these cells were found after exposure to ZnO-NPs, while alterations to a significant number of biological process and molecular pathways were found after exposure to TiO_2_-NPs.

### 3.1. Physical Properties of NPs

The physical characterizations of the NP batches showed how both NMs displayed the tendency to aggregation in the aqueous solution and to form a number of NPs populations whose sizes significantly exceeded the limit of 100 nm established to be considered an NM (Figure 2 and Figure 3). However, this tendency reduced in the cell culture medium, and the NP populations went below this limit of 100 nm during the exposure time of 72 h (Figure 2 and Figure 3). This fact could be associated with the presence of proteins in the culture medium that would help the dispersion of suspended particles by decreasing the number of aggregates, as previously reported by others [22,23,24]. Thus, the data shown in Figure 2 and Figure 3 suggest that all the effects herein reported could be attributed to materials on the nanometric scale; nevertheless, some degree of uncertainty remains as regards the real size of the nanoparticle population, since some interference of color of the cell culture medium with the laser wavelength could not be totally ruled out. It might also explain that the population of ZnO-NPs with the largest representation determined by TEM was 23 ± 8 nm (Table 1), while the population of ZnO-NPs with the largest representation in cell culture media determined by DLS was around 10 nm.

### 3.2. Uptake of NPs

It is proven by DLS cytometry that TiO_2_-NPs can be up-taken by certain cells lines as ARPE-19 (cultured human-derived retinal pigment epithelial cells) [25]. We demonstrated by both DLS cytometry (Table 2) and TEM (Figure 4) that, in our experimental conditions, the TiO_2_-NPs used in this work were able to be incorporated into the cytoplasm of T98G cells, while ZnO-NPs were unable to do so (Table 2 and Figure 4). We realized that ZnO-NPs incubated at a higher concentration might enter the cells because the Z-potential for both NPs are quite comparable (Table 1). Nevertheless, experiments for verifying this hypothesis are not technically feasible due to notable differences in cytotoxicity. This might explain the notable differences found between the transcriptomic alterations reported for both types of NPs. However, the inability of Zn-NPs to be incorporated into T98G cells alone cannot explain the lack of transcriptomic effects, because the silver NPs were unable to be incorporated into T98G cells as well but were, however, able to cause moderate transcriptomic alterations [15].

### 3.3. Cytotoxicity

It was noted that T98G cells were around four-fold more sensitive to ZnO-NPs than to TiO_2_-NPs, as the exposures to up to 20 µg/mL of TiO_2_-NPs and up to 5 µg/mL of ZnO-NPs for 72 h had no significant effect on cell survival (Table 3). This suggests that T98G human glioblastoma cells were considerably more resistant to TiO_2_-NPs than the U87 human astrocytoma cells, because the exposure of these cells to concentrations over 1 µg/mL for 48 h induced time and concentration reductions in cell survival [25]. However, the 48-h exposure of these U87 cells to 5 µg/mL of ZnO-NPs brought about only slight reductions in cell viability [26], which suggests that the sensitivity to ZnO-NPs may be similar to the sensitivity exhibited by T98G cells. The reason for these differences is unknown but can probably be attributed to the cellular model or to differences in the size of NPs, which were not well-clarified by Lai and coworkers [26]. Moreover, the primary astrocytes of neonatal Sprague-Dawley rats seem more sensitive than T98G cells to cytotoxicity induced by ZnO-NPs, as the exposure of these NPs to 8 µg/mL for 12 h induced a reduction in cell viability of around 50% [27].

### 3.4. Transcriptomic Alterations Induced by NPs

Surprisingly, T98G cells were much more susceptible to the transcriptomic alterations induced by TiO_2_-NPs than to those induced by ZnO-NPs. Indeed, no genes with an altered expression were identified in the RNAseq experiment after exposing T98G cells to the highest ZnO-NPs concentration, which caused cytotoxicity lower than 10%. Conversely, to the lack of alterations to the T98G cells induced by ZnO-NP exposure, certain cases are reported in the literature and show the effects of these NPs on glia cells. For example, Altunbek and coworkers [28] reported cytotoxicity and cycle perturbations in U373 human glioblastoma cells induced by pristine and ZnO-NP-bound albumin, fibrinogen and apo-transferrin (to the most abundant blood proteins). It was noted that ZnO-NPs were also able to induce apoptosis in the primary astrocytes of neonatal Sprague-Dawley rats via the phosphorylation of the c-Jun N-terminal kinase [27]. The alterations to the transcriptome induced by TiO_2_-NPs could be assigned to two of the main roles of glia cells, such as the maintaining homeostasis in the CNS through immunological processes and BBB integrity.

#### 3.4.1. Immunological Alterations

Several biological process and molecular functions that address the immunological processes, such as “*granulocyte chemotaxis*” (GO:0071621), “*response to lipopolysaccharide*” (GO:0032496), “*response to cytokine*” (GO:0034097) and “*cytokine activity*” (GO:0005125), were altered after exposure to TiO_2_-NPs (Table 4 and Table 5). Other molecular pathways, such as the “*interleukin signaling pathway*”, “*inflammation mediated by the chemokine and cytokine signaling pathways*”, “*B-cell activation*” and “*T-cell activation*”, were also altered (Table 6), which support these immunological alterations. The alterations to the immunological process were also confirmed in phenotypical terms by the ELISA determinations of the secreted IL8 (Table 9) and by the proteomic analysis, which found alterations to the biological process “*interleukin 12-mediated signaling pathway*” (GO:0035722) (Table 7) and a 2.3-fold overexpression of the protein macrophage migration inhibitory factor (Appendix A). This protein is a proinflammatory cytokine involved in innate immune responses to bacterial pathogens by playing the role of mediator in regulating the function of macrophages in host defenses [29]. All these phenotypic findings demonstrated that immunological impairments were not found due to a transcriptomic artefact.

Villeneuve and coworkers [30] established the adverse outcome pathway for the inflammatory response in a tissue. Initially, the tissue was under homeostatic conditions when it received the insult of inducers of inflammation (in our case, TiO_2_-NPs) and caused the activation of leukocytes (B- and T-cell activations were molecular pathways altered after TiO_2_-NPs; Table 6). Afterwards the recruitment of proinflammatory mediators, such as cytokines and interleukins, was expected, and we found that the response to cytokines, cytokine activity, the interleukin signaling pathway, the inflammation mediated by chemokine and the cytokine signaling pathway were altered after TiO_2_-NP exposure. Thus, the key events for inflammatory responses seemed to be met in all the found transcriptomic alterations. It would, therefore, be plausible that exposure to TiO_2_-NPs was able to induce neuroinflammation of the CNS. 

Inflammation as a response to NM exposure is an adverse outcome that has already been described both in vivo and in vitro. This is the case of the proinflammatory citoquine secretion found in microglia primary cultures or of the BBB inflammation of rats exposed to silicium dioxide NPs [13]. The overexpression of the genes involved in immune and inflammatory systems in the mouse brain has also been described after exposure to silver NPs [31]. Two TiO_2_-NPs and two silicon dioxides directly induced tumor necrosis factor α or IL6 secretion in the murine macrophage RAW 264.7 [32]. Conversely, to TiO_2_-NPs, we described a reduction in the secretion of IL6 and IL8 in T98G cells after exposure to silver NPs [15], which suggests that the response against inflammatory insults was also impaired in this case.

#### 3.4.2. Maintaining the BBB

The BBB is the physical barrier formed by endothelial tight junctions [33] that protects the CNS from toxins that blood might carry. Thus, the integrity and well-functioning of this barrier depends on the appropriate interactions among the endothelial cells forming it. Integrins are a large family of cell surface receptors that mediate cell adhesion [34] and should, therefore, play an important role in maintaining the BBB integrity. The transcriptomic analysis found alterations to the molecular function “*integrin binding*” (GO:0005178) (Table 5), the molecular pathway “*integrin signaling pathway*” (Table 6) and to the biological process of “*cell adhesion*” (GO:0007155) (Table 4). The biological process of “*cell–cell recognition*” (GO:0009988) was also altered in the proteomic analysis (Table 7), which supports the transcriptomic observations. The cadherin signaling pathway was also altered after TiO_2_ exposure (Table 6), which is another alteration that indicates a potential malfunction of the BBB, as cadherin complexes are critical for the assembly of cell–cell adhesion structures [35].

### 3.5. Autophagy

The TEM analysis (Figure 4) suggested that TiO_2_-NPs could induce autophagy in T98G cells, as confirmed by further RT-PCR experiments (Figure 6) that demonstrated a moderate activation of the LC3 and ATG3 genes 24 h after exposure. The LC3 gene encodes microtubule-associated protein 1 light chain 3 alpha, a protein required for autophagosome formation [36], while ATG3 encodes autophagy-related 3, which is also needed for autophagy [37]. The two-fold overexpression of protein sequestosome-1 (Appendix A) also supports the induction of autophagy after exposure to TiO_2_-NPs, since this protein is an autophagy receptor required for selective macroautophagy [38].

Unlike TiO_2_-NPs, no signs of autophagy were detected in the TEM micrographs of the cells exposed to ZnO-NPs (Figure 4). However, ZnO-NPs lower than 50 nm, but not within the micrometric range, are able to bring about the accumulation of autophagosomes and autophagic flux impairment in A549 lung epithelial cells [39]. As our ZnO-NPs fell within the same size range, it is plausible to speculate that the differences between TiO_2_- and ZnO-NPs can be due to intrinsic differences in cell lines or to a similar blockage to that reported for A549 cells.

### 3.6. Potential Anticancerigen Role of TiO_2_-NPs

The proteomic analysis revealed perturbations in the biological process related to the metabolization of anticancerigen drugs like doxorubicin (GO:0044598) and daunorubicin (GO:0044597) (Table 7). Both drugs belong to the anthracycline family, display similar structures and share metabolic routes. Anthracyclines are oxidized to unstable semi-quinone forms that return to parental anthracycline by releasing free radicals, which is the molecular mechanism of one of their pharmacological effects and is also the reason for their cardiotoxicity [40]. Thus, one of the forms of detoxication of anthracyclines is their reduction to avoid the semi-quinone formation. In this field, it has been shown that aldo-keto reductases C1 and C2 play a role in the metabolism and detoxification of these chemotherapy agents, while aldo-keto reductases C3 and B10 are overexpressed in certain resistant cell lines [41]. This scenario suggests that these aldo-keto reductases are also involved in anthracycline metabolism. Our proteomic analysis found a marked downregulation (from 55% to total suppression in the treated cells) of all four of these aldo-keto reductases (Appendix A), which suggests that TiO_2_-NPs might play a certain anticarcinogenic role themselves. Nevertheless, this should be further studied, because the MTT test detected no significant reduction in viability. 

It has been reported that neurological deficits associated with glioblastoma tumors are due to the fact that cancer glia cells develop, via WNT signaling pathways, a network of microtubes that wrap neurons to cause neurodegeneration [42]. Table 6 shows how the Wnt signaling pathway is significantly downregulated after exposure to TiO_2_-NPs because, for example, the Wnt 6 ligand is under-expressed by around 42%, as are other proteins in the pathway, such as the nuclear factor of activated T cells, cytoplasmic 4 (under-expressed by 52%), protocadherin-18 (under-expressed by 63%) and cadherin-11 (under-expressed by 54%). This suggests that the TiO_2_-NP-exposed T98G cells may have a malignancy route that is not fully operative compared to the unexposed cells, which could confer onto TiO_2_-NPs certain therapeutic properties that still need to be further investigated.

Angiogenesis seemed affected by the exposure to TiO_2_-NPs through the repression of the expression of genes c-Fos and Ephrin-B1 (Table 6). Both genes promote vascular and epithelial angiogenesis development [43,44], and, therefore, repression might contribute to diminishing the capability of glioblastomas to extend via angiogenesis. Nevertheless, the real capability of TiO_2_-NPs to interfere with angiogenesis should be further investigated phenotypically. Likewise, autophagy can play a double role in tumors, as it protects cancer cells from chemotherapeutics but can also kill cells in which apoptosis pathways are inactive [45], which is the case of most cancer cells. The role of TiO_2_-NP-induced autophagy should be further investigated to clarify whether it acts as a protective or an anticancerigen factor.

## 4. Materials and Methods

### 4.1. Nanoparticles

TiO_2_-NPs with an average diameter of 27 ± 7 nm (catalog No. TIO-00, batch 17001) and ZnO-NPs with an average diameter of 20 ± 4 nm (catalog No. ZNO-00, batch 17001) were provided by SCHARLAB (Madrid, Spain) (www.scharlab.com, accessed on 29 December 2020) as white solids.

For both NPs, stock solutions (400 µg/mL) were prepared in phosphate-buffered saline (PBS). These stock solutions were sonicated for 10 min at 110 watts immediately before taking an appropriate volume to prepare NP suspensions at appropriate concentrations in fresh culture medium. The fresh cellular media with NP suspension were sonicated again for 10 min at 110 watts immediately before cellular exposure began.

### 4.2. Physicochemical Characterisation of Nanoparticles

The supplier provided the TEM images shown in Appendix A and the distribution sizes provided in Appendix A. TiO_2_-NPs displayed a maximum number of particles between 20 nm and 25 nm (Appendix A), while the maximum number of particles lay between 16 nm and 18 nm for ZnO-NPs (Appendix A). We requested an additional physical characterization of the nanoparticles batch from Nanoimmunotech SL (https://nanoimmunotech.eu/en, accessed on 29 December 2020), which included the parameters described below.

#### 4.2.1. Particle Size in Water by Dynamic Light Scattering

NPs were diluted in type 1 water (18 MΩ.cm) with 25 µg/mL of Zn-NPs and 6.25 µg/mL of TiO_2_-NPs. They were further analyzed in quintuplicate with a minimum of 10 runs per measurement using Malvern Zetasizer ZS equipment at 25 °C.

#### 4.2.2. Z-Potential in Water

NPs were diluted in type 1 water (18 MΩ.cm) and were analyzed in triplicate after adjusting the number of runs to the specific necessity of each sample using Malvern Zetasizer ZS equipment at 25 °C.

#### 4.2.3. Transmission Electron Microscopy

The air-dried samples were placed on copper grids with a carbon film for further analyses done using a TECNAI F30 (300 kV) microscope. Images were processed with ImageJ free software.

#### 4.2.4. Stability of NPs in the Cell Culture Medium

Suspensions of NPs were prepared in the Roswell Park Memorial Institute (RPMI) cellular medium supplemented with 10% fetal bovine serum (FBS) and type I water. These NPs suspensions were incubated at 37 °C for up to 72 h. Particle size was determined by DLS, as described above.

### 4.3. Cellular Cultures

T98G human glioblastoma cells were provided by the European Collection of Authenticated Cell Cultures (catalog No. 92090213) (https://www.phe-culturecollections.org.uk/collections/ecacc.aspx, accessed on 29 December 2020). Cells were cultured at 37 °C in a 5% CO_2_ atmosphere using the cellular media recommended by the supplier, as described by Fuster and coworkers [15]. Cells were expanded in P100 TPP plates until confluence before being trypsinized, as previously described [15], and sub-cultured in either P60 TPP plates or 96-well trays at the appropriate density according to the specific experiment.

T98G human glioblastoma cells were obtained from a multiform tumor found in a 61-year-old Caucasian male and behaved as normal glial cells, except the cellular cycle was arrested in the G1 phase. Our lab performed a comparison of the transcriptome of normal glial cells and T98G cells, finding differential expressions in only 4875 genes, but no cellular pathway was under- or overrepresented among these 4875 genes when compared with the 24,628 expressed genes [15]. It suggests that T98G cells can be used for testing the neurotoxicity of NMs, since no drastic differences in the capability of the response to xenobiotics between T98G cells and glial cells were expected.

### 4.4. Cellular Exposure to Nanoparticles

Cells were seeded in suitable trays at appropriate densities on experiment day 0 and were incubated in complete medium at 37 °C and in a 5% CO_2_ atmosphere for 24 h. On day 1, the cell culture medium was removed. Afterwards, the fresh culture medium, containing appropriate concentrations of TiO_2_-NPs or Zn-NPs, was added to cells, and an appropriate exposure time was allowed (72 h, in most cases).

### 4.5. Cellular Viability Tests

Cellular viability after exposure to TiO_2_-NPs and Zn-NPs was assessed by the 3-(4,5-dimethylthiazol-2-yl)-2,5-diphenyltetrazoliumbromide (MTT) test, as described by Fuster and coworkers [15]. All the experimental conditions were assayed (for both tests) during each independent experiment with six biological replicates (6 different wells of the same culture on the same tray). Between four and six independent experiments for each experimental condition were performed. In all cases, the percentage of viability of the cell cultures exposed to both NPs was estimated by assuming the absorbance of the control cultures to be 100% viability (not exposed to NPs). In all the experiments, the cytotoxicity-positive controls were run using 10 µg/mL of CuSO_4_, which was able to reduce the viability of the exposed cultures by around 80% (data not shown).

### 4.6. Flow Cytometry

Flow cytometry services were provided by ACTI (https://www.um.es/web/acti/, accessed on 29 December 2020) (University of Murcia, Murcia, Spain). A FACS Canto flow cytometer (Becton Dickinson) equipped with an Ar laser of 488 nm and 20 mV and a He laser of 633 nm and 17 mV was employed. T98G human glioblastoma cells were exposed in P60 TPP plates (300,000 cells/plate seeded on day 0) to either 20 µg/mL of TiO_2_-NPs or 5 µg/mL of ZnO-NPs for 72 h, as described in Section 4.4. Afterwards, NPs were removed, and cells were washed with PBS and detached from plates with trypsin to be further processed for light scattering cytometry or cell viability, as described by Fuster and coworkers [15]. In short, for light scattering cytometry, cells were passed through the cytometer, which was set up to logarithmically measure the small-angle forward scatter (FSC) intensity (approx. 0–5°) and SSC intensity (approx. 85–95°) of 20,000 events. The viability of the cell cultures was assessed with propidium iodide.

### 4.7. Electron Microscopy

Electron microscopy services were provided by ACTI. T98G human glioblastoma cells were exposed in P60 TPP plates (300,000 cells/plate seeded on day 0) to either 20 µg/mL of TiO_2_-NPs or 5 µg/mL of ZnO-NP for 72 h, as described in Section 4.4. Afterwards, NPs were removed, and cells were washed with PBS before being detached from plates with trypsin, as described by Fuster and coworkers [15]. At the end of exposure, NPs were removed, and cells were pelleted and resuspended with 100 µL of 3% glutaraldehyde in 0.1-M cacodylate buffer (pH 7.4) for fixation purposes. Cells were maintained in this fixative solution for 5 min at room temperature, followed by 1 h at 4 °C before removing the fixative solution and resuspending cells with 8% sucrose in 0.1-M cacodylate buffer, pH 7.4. Cells were then carried in an ice bath to the ACTI facilities, where they were treated as described by Fuster and coworkers [15], depending on the scope of the experiment. One independent experiment and two others (each one with two biological replicates) were performed for the TEM and SEM analyses, respectively.

### 4.8. RNA Isolation

The T98G human glioblastoma cells seeded on day 0 as 160,000 cells/plate in P60 TPP plates were exposed to either 20 µg/mL of TiO_2_-NPs or 5 µg/mL of ZnO-NPs for 72 h, as described in Section 2.4. At the end of the 72-h exposure period, NPs were removed, and RNA was isolated using the Trizol reagent, as previously described by Fuster and coworkers [15].

### 4.9. Massive Parallel RNA Sequencing

Three biological replicates per experimental condition of the T98G human glioblastoma cell cultures seeded on day 0 as 160,000 cells/plate in P60 TPP plates were exposed to either 20 µg/mL of TiO_2_-NPs or 5 µg/mL of ZnO-NPS for 72 h, as described in Section 4.4. These doses were chosen as the maximal doses able to reduce the cell viability by no more than 10% (see results of the cell viability tests). Afterwards, the RNA of the resulting cultures was isolated as described in Section 4.8. Only the samples with an RNA integrity number (RIN) above 7 were considered appropriate for the massive parallel RNA sequencing (RNAseq) experiments, which were performed by MACROGEN Inc. (https://dna.macrogen.com/eng/, accessed on 29 December 2020) on the Illumina platform using paired-end 101-bp reads. The RNAseq raw data were deposited in the National Center for Biotechnology Information (NCBI) (Bethesda, USA) Sequence Read Archive (SRA) (https://www.ncbi.nlm.nih.gov/sra/, accessed on 29 December 2020) with Bio Project accession number PRJNA580150.

### 4.10. Bioinformatic Analysis

The bioinformatic analysis was performed as reported by Fuster and coworkers [15]. This analysis used the following software: (i) Trimmomatic v. 0.36 for trimming low-quality bases and any remaining adapter sequences present in the reads, (ii) Hisat2 v. 2.1.0 for mapping the read pairs to the GRCh38 version of the human reference genome, (iii) Samtools v. 0.1.19 for compressing the resulting SAM files into BAM files and (iv) Cufflinks v. 2.2.1 for quantifying gene expression levels and making statistical comparisons.

The PANTHER (Protein ANalysis THrough Evolutionary Relationships) v. 13.1 Classification System (http://pantherdb.org/, accessed on 29 December 2020) was used to assign Gene Ontology (GO) (http://www.geneontology.org/, accessed on 29 December 2020) terms to the differentially expressed genes and to determine the nature of their products and the pathways in which they participated. PANTHER allows the analysis of transcriptomes combining gene function, ontology, pathways and statistical analysis tools [46].

### 4.11. Validation of Massive Parallel RNA Sequencing by Real-Time PCR

The expression levels of selected genes identified in the RNAseq experiment were further validated in independent experiments by RT-PCR using the Step One Plus Real-Time PCR System (Applied Biosystems, Madrid, Spain). For this purpose, the T98G human glioblastoma cells seeded on day 0 as 160,000 cells/plate in P60 TPP plates were exposed to 20 µg/mL TiO_2_-NPs, as described in Section 4.4. The expression levels of the target genes were further assayed in a process divided into three steps: (i) RNA isolation (as described in Section 4.8), (ii) RNA reverse transcription and (iii) gene expression quantification using TaqMan® kits. The housekeeping genes for the gene expression quantification were selected using the TaqMan™ Array Human Endogenous Controls Plate (Applied Biosystem catalog No. 4426696). RNA reverse transcription and gene expression quantification were performed as previously described by Fuster and coworkers [15].

### 4.12. Proteomic Determinations

Proteomic services were provided by ACTI. T98G human glioblastoma cells were exposed in P60 TPP plates (160,000 cells/plate seeded on day 0) to 20 µg/mL of TiO_2_-NPs for 72 h, as described in Section 4.4. Titanium dioxide NPs were then removed, and cells were washed with PBS before being detached from plates with trypsin. Afterwards, the cytoplasmic and membrane protein fractions were separated using the Thermo Scientific Subcellular Protein Fractionation Kit for Cultured Cells (catalog No. 78840) following the supplier’s instructions. Finally, the total amount of protein was determined using the Thermo Pierce BCA Protein Assay Kit (catalog No. 23227) following the supplier’s instructions. Around 100 µg of protein of each fraction were sent to ACTI for the proteomic determinations following the procedures described below. Each experimental condition was assayed in three biological replicates.

#### 4.12.1. Trypsin Digestion

Samples (approx. 100 µg) were dissolved in 200 µL of 50-mM ammonium bicarbonate buffer, pH 8.5, with 0.01% ProteaseMax (Promega) to enhance trypsin digestion. Protein samples were reduced by adding 10-mM DTT at 56 °C for 20 min. Then, samples were alkylated by adding 25-mM iodoacetamide for 30 min at room temperature in the dark. Digestion was performed by adding one µg of Trypsin Gold Proteomics Grade (Promega) (approximately 1:100, *w/w*) for 3 h at 37 °C. The reaction was stopped with 0.1% formic acid and filtered through a 0.2-µm mesh. Finally, samples were dried in an Eppendorf 5301 Vacuum Concentrator model. 

#### 4.12.2. HPLC-MS/MS Analysis

The separation and analysis of the tryptic digests of the samples were performed by a HPLC/MS system consisting of an Agilent 1290 Infinity II Series HPLC (Agilent Technologies, Santa Clara, CA, USA), equipped with an Agilent 6550 Quadrupole Time of Flight (Q-TOF) Mass Spectrometer (Agilent Technologies, Santa Clara, CA, USA) using an Agilent Jet Stream Dual electrospray (AJS-Dual Electrospray ionization (ESI)) interface. The experimental parameters for HPLC and Q-TOF were set in the MassHunter Workstation Data Acquisition software (Agilent Technologies, Rev. B.08.00).

The dry samples from trypsin digestion were resuspended in 20 µL of buffer A, which consisted of water/acetonitrile/formic acid (94.9:5:0.1). Samples were injected into an Agilent AdvanceBio Peptide Mapping HPLC column (2.7 µm, 100 × 2.1 mm, Agilent Technologies) and thermostatted at 55 °C at a flow rate of 0.4 mL/min. This column is suitable for the separation and analysis of peptides. After injections, the column was washed with buffer A for 2 min, and the digested peptides were eluted using linear gradient 0–40% B (buffer B: water/acetonitrile/formic acid, 10:89.9:0.1) for 30 min. 

The mass spectrometer operated in the positive mode. The nebulizer gas pressure, drying gas flow and seat gas flow were, respectively, set at 35 psi, 14 l/min at 300 °C and 11 l/min at 250 °C. The capillary spray, fragmentor and octopole RF Vpp voltages were 3500 V, 360 V and 750 V, respectively. The profile data were acquired for both the MS and MS/MS scans in the extended dynamic range mode. The MS and MS/MS mass ranges were 50–1700 m/z, and the scan rates were 8 spectra/sec for MS and 3 spectra/sec for MS/MS. The auto-MS/MS mode was used with precursor selection by abundance and a maximum of 20 precursors selected per cycle. Ramp collision energy was employed with a slope of 3.6 and an offset of −4.8. The same ion was rejected after two consecutive scans.

Data processing and analysis were performed with the Spectrum Mill MS Proteomics Workbench (Rev B.06.00.201, Agilent Technologies, Santa Clara, CA, USA). The MS/MS search against the appropriate and updated protein database (UniProt, human proteins, 2018/11/29) was performed according to these criteria: variable modifications search mode (carbamidomethylated cysteines, STY phosphorylation, oxidized methionine and N-terminal glutamine conversion into pyroglutamic acid); tryptic digestion with five maximum missed cleavages; ESI-Q-TOF instrument; minimum matched peak intensity 50%; maximum ambiguous precursor charge +5; monoisotopic masses; peptide precursor mass tolerance 20 ppm; product ion mass tolerance 50 ppm and the calculation of reversed database scores. Validation of the peptide and protein data was performed using auto-thresholds.

### 4.13. Interleukin Determinations

The concentrations of interleukin (IL) 8 (IL-8) in the cellular medium were determined by ELISA procedures using the Human IL 8 ELISA Kit (ThermoFisher Scientific (Madrid, Spain) catalog No. KHC0081) following the supplier’s instructions (Invitrogen). All the experimental conditions were assayed during each independent experiment with three different wells of the same culture on the same tray (three biological replicates). IL8 was determined after exposures lasting 24 h and 72 h to TiO_2_- and ZnO-NPs. The 72-h exposure involved three independent experiments, with two more for the 24-h exposure. The lipopolysaccharides from *Escherichia coli* O55:B5 (MERK, catalog No. L6529) (300 µg/mL) were used in all the experiments as a positive control.

## 5. Conclusions

This work reported how exposing T98G human glioblastoma cells to TiO_2_-NPs induces a potential disruption in the BBB integrity and neuroinflammation. Both issues might be relevant from a toxicological point of view. On the one hand, BBB disruption may cause neurotoxicity, because it would allow a greater bioavailability of other toxicants circulating in the blood. On the other hand, inflammation is indeed a mechanism of defense of the organism against external insults. However, neuroinflammation has been included in the adverse outcome pathway for parkinsonian motor deficits, as this neuroinflammation causes the degeneration of dopaminergic neurons [47]. Therefore, the results of this work suggest a plausible mechanistic link between chronic exposure to TiO_2_-NPs and the further development of neurodegenerative diseases.

The potential capability of using TiO_2_-NPs in the treatment of glioblastomas and other tumor types is worthy of attention, because transcriptomic and proteomic analyses suggest that these NPs themselves might reduce the resistance of tumor cells to the anticancer pharmaceuticals belonging to anthracycline family and may reduce the capability of metastasis by blocking angiogenesis. These properties would make these NPs very promising agents against cancer, especially if they can be combined with other chemotherapy agents. 

Other authors have estimated that the risk for the general (nonoccupationally exposed) population to develop adverse effects by TiO_2_-NP exposure cannot be excluded [48], especially for the liver and reproduction system [49]. Other studies have also revealed the tendency of TiO_2_-NPs to accumulate in rat brains after oral exposure [11]. These findings suggest that reassessing the risk associated with TiO_2_-NPs would be necessary by also using the CNS as a target organ, especially after studies have revealed that exposures are reaching relatively high values, e.g., 0.74-, 1.61- and 4.16-mg/kg body weight (bw)/day, respectively, for the elderly, 7–69-year-old people and young children as the 95 percentile values [50]. 

## Figures and Tables

**Figure 1 ijms-22-02084-f001:**
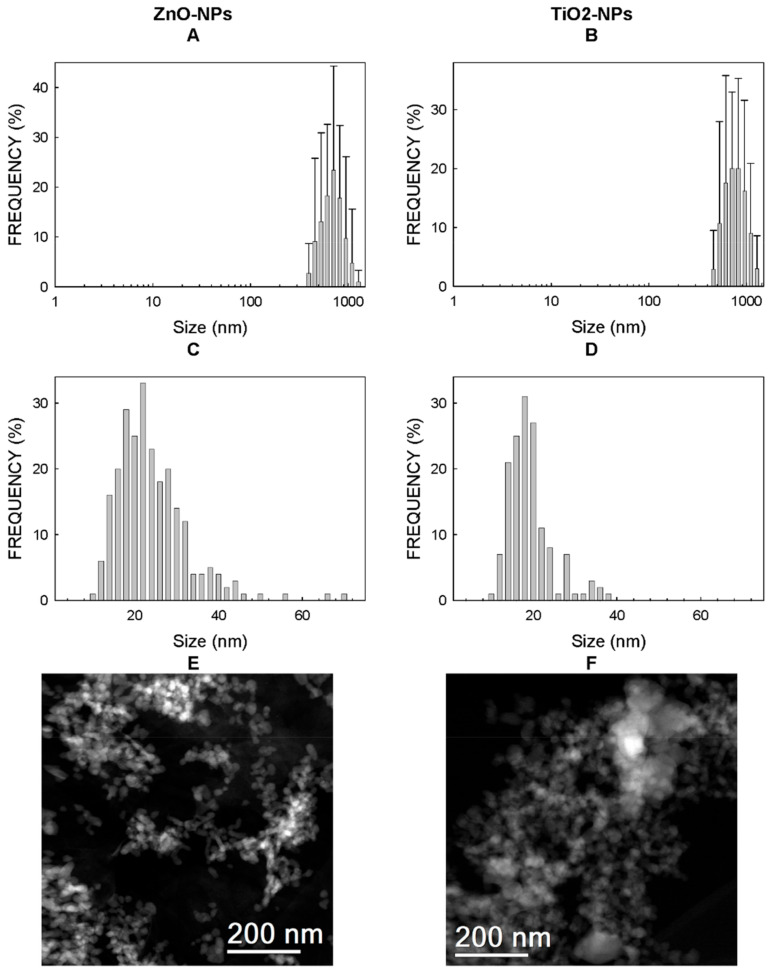
Physicochemical properties of nanoparticles (NPs). Histograms with the distribution of the sizes calculated by dynamic light scattering (DLS) (**A**,**B**) and transmission electron microscopy (TEM) (**C**,**D**). Pictures of both NPs for TEM (**E**,**F**).

**Figure 2 ijms-22-02084-f002:**
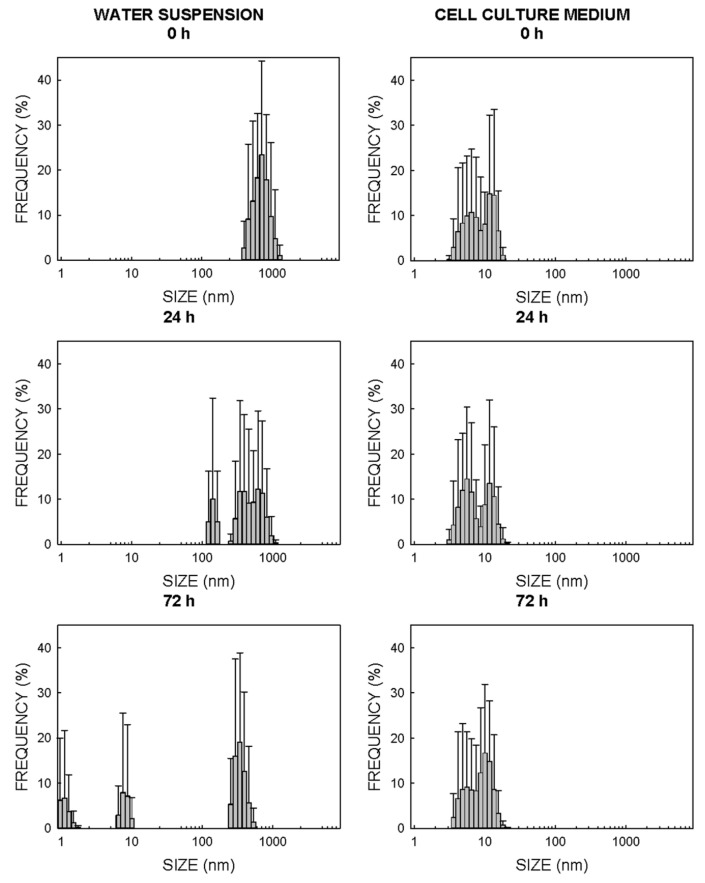
Analysis of the size and distribution of ZnO-NPs in water (**left panels**) and the cell culture medium (**right panels**). Size was determined by DLS. The cell culture medium was the Roswell Park Memorial Institute (RPMI) cellular medium supplemented with 10% fetal bovine serum (FBS).

**Figure 3 ijms-22-02084-f003:**
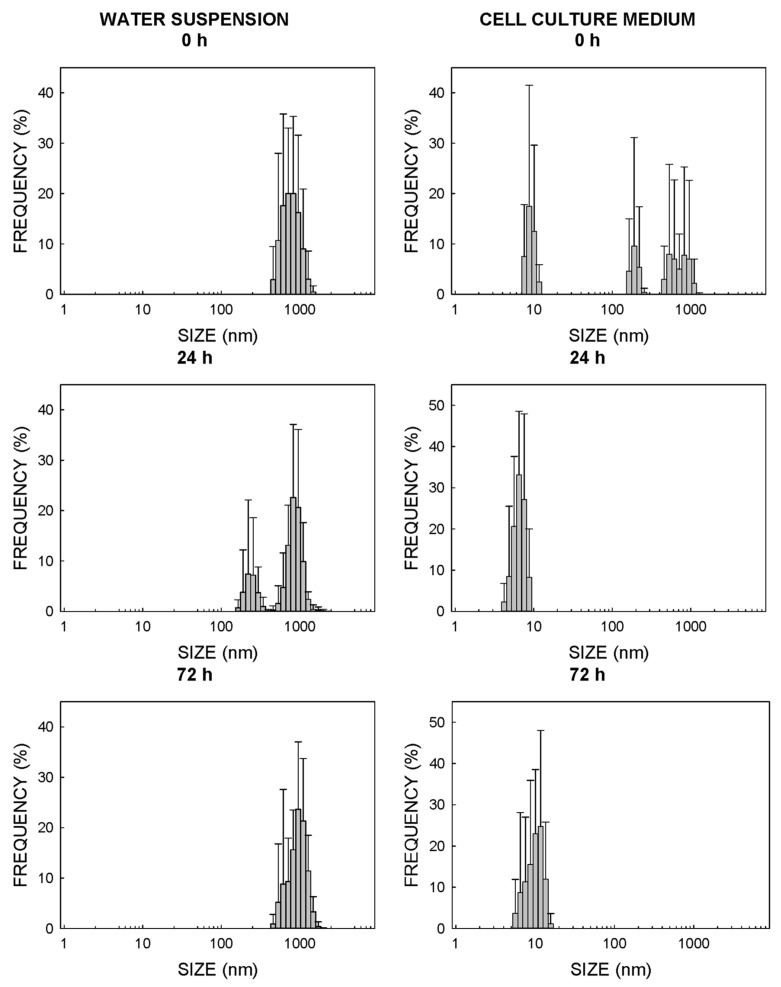
Analysis of the size and distribution of TiO_2_-NPs in water (**left panels**) and the cell culture medium (**right panels**). Size was determined by DLS. The cell culture medium was the the Roswell Park Memorial Institute (RPMI) cellular medium supplemented with 10% FBS.

**Figure 4 ijms-22-02084-f004:**
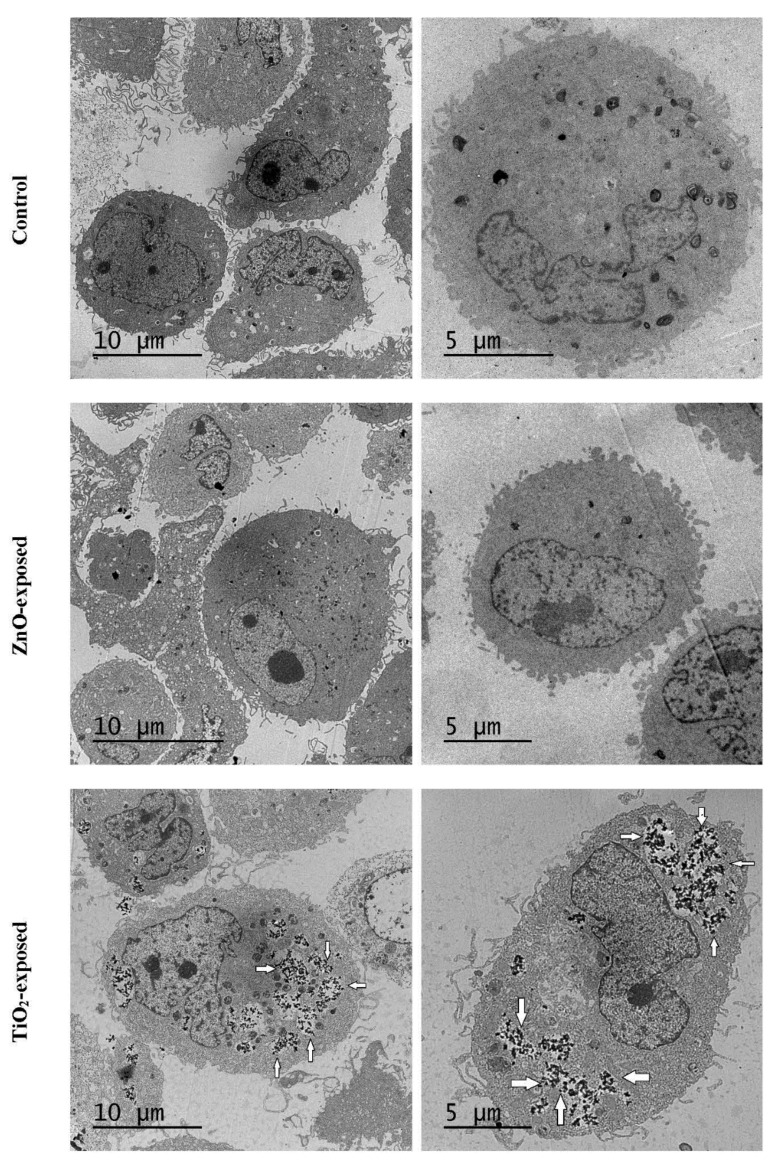
T98G human glioblastoma cells visualized by TEM. Cells were exposed to 20 µg/mL of TiO_2_-NPs or 5 µg ZnO-NPs for 72 h, as described in Section 4.4, and then prepared for TEM as described in Section 4.7. A second independent culture yielded similar results. White arrows denote clusters of TiO_2_-NPs in the areas of apparent cytoplasmic autophagy.

**Figure 5 ijms-22-02084-f005:**
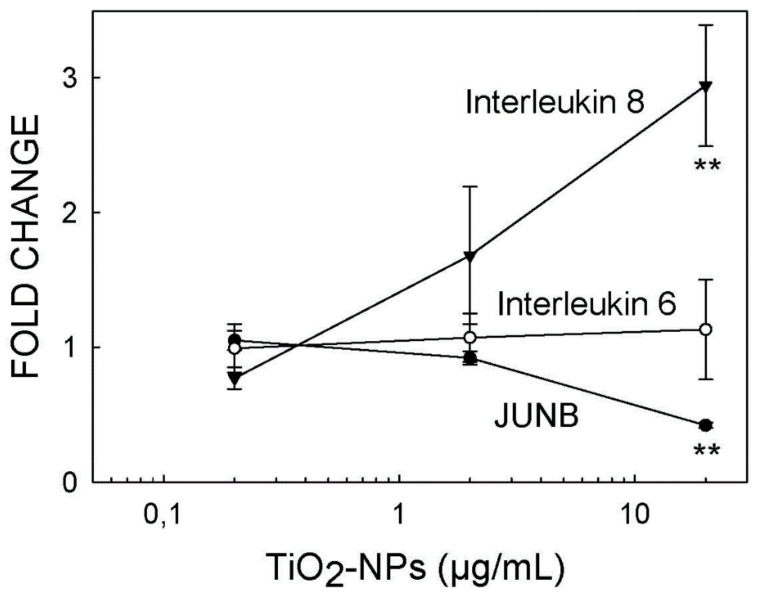
Effect of the concentration on alterations to the gene expressions induced by TiO_2_-NPs. The mean ± standard deviations of three biological replicates are displayed. ** = statistically different from the control for *p* < 0.01.

**Figure 6 ijms-22-02084-f006:**
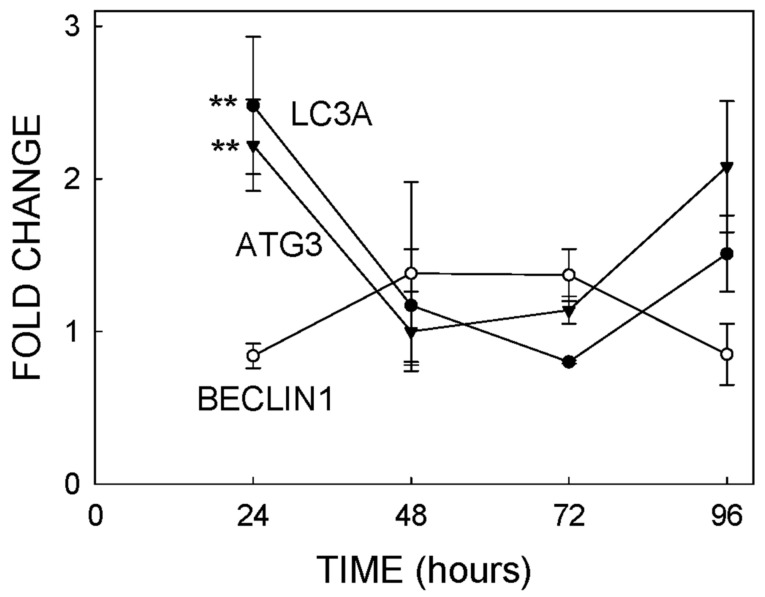
Expression of three different genes involved in cellular autophagy. Cells were exposed to 20 µg/mL of TiO_2_-NPs, and the mRNA levels of three genes were determined by PCR, as described in Section 4.11. Each experimental condition was assayed with three biological replicates. ** = expression statistically differed from the control for *p* < 0.01. Microtubule-associated protein 1 light chain 3 alpha = LC3A and autophagy-related 3 = ATG3.

**Table 1 ijms-22-02084-t001:** Physicochemical properties of the TiO_2_- and ZnO-NPs (nanoparticles). Figure 1 shows the histograms with the distribution of the sizes determined by both dynamic light scattering (DLS) and transmission electron microscopy (TEM). Appendix A shows the three Z-potential determinations run per each nanomaterial (NM).

Physical Parameter	ZnO-NPs	TiO_2_-NP
Mean size by DLS (nm)	707.9 ± 173.3	786.9 ± 176.7
Mean size by TEM (nm)	23.15 ± 8.65	18.18 ± 5.25
Z-potential (mV)	+17.0 ± 0.6	+22.8 ± 0.8

**Table 2 ijms-22-02084-t002:** Mean side scatter (SSC) and viability of the cell cultures exposed to NPs. Cells were exposed to 20 µg/mL of TiO_2_-NPs or 5 µg/mL of ZnO-NPs, as described in Section 4.4. Then, they were treated as described in Section 4.6 for flow cytometry. Two biological replicates were used for each experiment. Here we see the individual records of the biological replicates and their respective means (in brackets). Viability was determined using propidium iodide only in the second experiment. Ratios were estimated with the means as the SSC sample/SSC control. These experiments were run in parallel to the silver NPs and shared controls with the data published in [15].

Nanoparticle	Experiment 1 (72 h)		Experiment 2 (24 h)			Experiment 3 (72 h)		
	SSC	Ratio	SSC	Ratio	Viability (%)	SSC	Ratio	Viability (%)
Control	1168		1038		98	816		98
	1186		979		98	680		96
	(1177)	1	(1009)	1	(98)	(748)	1.00	(97)
ZnO-NPs	1192		1133		97	1003		92
	1281		1122		98	1031		96
	1237	1.05	(1128)	1.12	(98)	(1017)	1.36	(94)
TiO_2_-NP	2000		3904		90	5327		92
	2244		4059		92	4786		91
	(2122)	1.8	(3982)	(3.95)	(91)	(5057)	6.76	(92)

**Table 3 ijms-22-02084-t003:** Effect of the model NPs on the cell viability of T98G human glioblastoma cells. Human neuroblastoma T98G cells were exposed to NPs, as defined in Section 4.4 of the Material and Methods. Afterwards, substances were removed, and the viability of the cultures was determined by the 3-(4,5-dimethylthiazol-2-yl)-2,5-diphenyltetrazoliumbromide (MTT) test. The mean ± standard deviation of the percentage of viability for *n* independent experiments is displayed. Each independent experiment was performed with six wells. Shadowed cells show the concentration taken as the maximum tolerable dose, which was further used for the RNAseq experiments.

Nanoparticle	Concentration (µg/mL)	% of Viability (*n*)
ZnO-NPs	10	87 ± 1 (*n* = 5)
	5	92 ± 1 (*n* = 4)
TiO_2_-NPs	30	87 ± 5 (*n* = 4)
	20	90 ± 3 (*n* = 6)
	15	92 ± 2 (*n* = 6)
	10	93 ± 1 (*n* = 6)

**Table 4 ijms-22-02084-t004:** The slim biological process ontology terms overrepresented in the differentially expressed genes in the T98G human glioblastoma exposed for 72 h to 20 µg/mL of TiO_2_-NPs. Only the results for FDR *p* < 0.05 are displayed. DEG = differentially expressed genes, EF = enrichment factor, GO = Gene Ontology and FDR = false discovery rate.

Biological Process Sub-Ontology (GO Term)	Genes in Reference List	Genes among DEG	Expected among DEG	EF	*p*-Value	FDR
Granulocyte chemotaxis (GO:0071621)	8	3	0.14	22	7.26 × 10^−4^	4.98 × 10^−2^
Response to lipopolysaccharide (GO:0032496)	13	4	0.23	18	1.63 × 10^−4^	1.87 × 10^−2^
Positive regulation of multicellular organismal process (GO:0051240)	18	4	0.31	13	4.69 × 10^−4^	3.66 × 10^−2^
Response to cytokine (GO:0034097)	39	6	0.68	9	1.06 × 10^−4^	1.51 × 10^−2^
Regulation of cell differentiation (GO:0045595)	50	7	0.87	8	4.80 × 10^−5^	8.23 × 10^−3^
Transmembrane receptor protein tyrosine kinase signaling pathway (GO:0007169)	89	8	1.54	5	2.43 × 10^−4^	2.31 × 10^−2^
Cell adhesion (GO:0007155)	146	12	2.53	5	1.59 × 10^−5^	1.37 × 10^−2^
Generation of neurons (GO:0048699)	122	9	2.11	4	3.96 × 10^−4^	3.24 × 10^−2^

**Table 5 ijms-22-02084-t005:** The slim molecular function ontology terms overrepresented in the differentially expressed genes in the T98G human glioblastoma exposed for 72 h to 20 µg/mL of TiO_2_-NPs. Only the results for FDR *p* < 0.05 are displayed.

Molecular Function Sub-Ontology (GO Term)	Genes in Reference List	Genes among DEG	Expected among DEG	EF	*p*-Value	FDR
Integrin binding (GO:0005178)	14	4	0.24	16	2.07 × 10^−4^	3.42 × 10^−2^
Cytokine activity (GO:0005125)	32	5	0.55	9	3.81 × 10^−4^	2.70 × 10^−2^
RNA polymerase II proximal promoter sequence-specific DNA binding (GO:0000978)	71	7	1.23	6	3.55 × 10^−4^	2.93 × 10^−2^
Signaling receptor activity (GO:0038023)	231	13	4.00	3	2.75 × 10^−4^	3.42 × 10^−2^

**Table 6 ijms-22-02084-t006:** Molecular pathways altered in the T98G human glioblastoma exposed for 72 h to 20 µg/mL of TiO_2_-NPs.

Pathway	Genes Involved	Log_2_ Fold Change
Angiogenesis	Ephrin-B1	−1.08
	Proto-oncogene c-Fos	−1.68
TGF-beta signaling pathway	Transcription factor JUN-B	−1.26
	Inhibin beta A chain	1.14
Heterotrimeric G-protein signaling pathway and Gi alpha and Gs alpha mediated pathway	Regulator of G-protein signaling	−1.36
	Alpha-1D adrenergic receptor	−1.38
	Adenosine receptor A1	−1.71
Blood coagulation	Tissue-type plasminogen activator	1.17
	Integrin beta 3	1.59
p53 pathway	Thrombospondin-1	−1.02
	Ribonucleoside-diphosphate reductase subunit M2	−1.13
Cadherin signaling pathway	Protein Wnt-6	−1.14
	Protocadherin-18	−1.43
	Cadherin-11	−1.13
Interleukin signaling pathway	Interleukin 8	1.51
	Proto-oncogene c-Fos	−1.68
Integrin signaling pathway	Integrin beta 8	−1.32
	Integrin alpha X	1.06
	Rho-related GTP-binding protein RhoE	1.03
	Laminin subunit alpha-4	−1.14
	Integrin beta 3	1.59
CCKR signaling map	Early growth response protein 1	−2.48
	Transcription factor 4	−1.20
	Regulator of G−protein signaling 2	−1.36
	Interleukin-8	1.51
	Proto-oncogene c−Fos	−1.68
Inflammation mediated by chemokine and cytokine signaling pathway	1-phosphatidylinositol 4,5-bisphosphate phosphodiesterase delta-4	−1.18
	Nuclear factor of activated T-cells, cytoplasmic 4	−1.07
	C3a anaphylatoxin chemotactic receptor	1.33
	Interleukin 8	1.51
	Transcription factor JUN-B	−1.26
	C-C motif chemokine 2	−1.72
EGF receptor signaling pathway	Pro-neuregulin-1, membrane-bound isoform	1.25
	Protein sprouty homolog 4	−1.54
Heterotrimeric G-protein signaling pathway-Gq alpha and Go alpha-mediated pathway	Regulator of G-protein signaling 2	−1.36
	Adenosine receptor A1	−1.71
Parkinson’s disease	Heat shock 70-kDa protein 1A	1.33
	Heat shock 70-kDa protein 1B	1.29
	Synphilin-1	2.07
B-cell activation	Nuclear factor of activated T-cells, cytoplasmic 4	−1.07
	B-cell receptor CD22	1.84
	Proto-oncogene c-Fos	−1.68
Gonadotropin releasing hormone receptor pathway	Nuclear factor of activated T-cells, cytoplasmic 4	−1.07
	Early growth response protein 1	−2.48
	Heat shock 70-kDa protein 1A	1.33
	Transcription factor JUN-B	−1.26
	Heat shock 70-kDa protein 1B	1.29
	Muellerian-inhibiting factor	−1.01
	Inhibin beta A chain 08476	1.14
	Proto-oncogene c-Fos	−1.68
	Cyclic AMP-dependent transcription factor ATF-3	1.11
Apoptosis signaling pathway	Heat shock-70 kDa protein 1A	1.33
	Heat shock-70 kDa protein 1B	1.29
	Proto-oncogene c-Fos	−1.68
	Cyclic AMP-dependent transcription factor ATF-3	1.11
Wnt signaling pathway	Nuclear factor of activated T-cells, cytoplasmic 4	−1.07
	Protein Wnt-6	−1.14
	Protocadherin-18	−1.43
	Cadherin-11	−1.13
T cell activation	HLA class II histocompatibility antigen gamma chain	−1.43
	Nuclear factor of activated T-cells, cytoplasmic 4	−1.07
	Proto-oncogene c-Fos	−1.68

**Table 7 ijms-22-02084-t007:** The biological process ontology terms overrepresented in the differentially expressed proteins in the T98G human glioblastoma exposed for 72 h to 20 µg/mL of TiO_2_-NPs. Only the results for FDR *p* < 0.05 are displayed. DEP = differentially expressed portions.

Biological Process Sub-Ontology (GO Term)	Proteins in Reference List	Proteins among DEP	Expected among DEP	EF	*p*-Value	FDR
Doxorubicin metabolic process (GO:0044598)	9	2	0.01	>100	1.94 × 10^−5^	2.57 × 10^−25^
Daunorubicin metabolic process (GO:0044597)	9	2	0.01	>100	1.94 × 10^−5^	2.37 × 10^−2^
Interleukin 12-mediated signaling pathway (GO:0035722)	46	3	0.03	>100	3.35 × 10^−9^	1.78 × 10^−2^
Cell-cell recognition (GO:0009988)	69	3	0.04	70.2	1.08 × 10^−5^	1.90 × 10^−2^
Protein folding (GO:0006457)	220	4	0.14	29	8.34 × 10^−6^	1.66 × 10^−2^
Symbiotic process (GO:0044403)	783	5	0.48	10	7.34 × 10^−5^	4.49 × 10^−2^
Organic substance catabolic process (GO:1901575)	1771	7	1.10	6.4	3.34 × 10^−5^	2.65 × 10^−2^
Regulation of biological quality (GO:0065008)	4073	11	2.52	4.4	7.82 × 10^−7^	6.21 × 10^−3^
Transport (GO:0006810)	4550	10	2.82	3.5	3.42 × 10^−5^	2.59 × 10^−2^

**Table 8 ijms-22-02084-t008:** Effects of NPs on the expressions of the different genes. T98G human glioblastoma cells were exposed to 20 µg/mL of TiO_2_-NPs for 72 h, as described in Section 4.4. Then, RNA was isolated as described in Section 4.8 to be reverse-transcribed and quantified by real-time quantitative polymerase chain reactions (RT-PCR), as described in Section 4.11, using glyceraldehyde 3-phosphate dehydrogenase (GAPDH) and phosphoglycerate kinase 1 (PGK1) as the housekeeping genes. The RNA sequencing (RNAseq) data were taken from the respective log_2_ (fold change (FC)), as shown in Appendix A. * = statistically different from the control for *p* < 0.05. ** = statistically different from the control for *p* < 0.01. *** = statistically different from the control for *p* < 0.001 IL = interleukin.

		Relative Expression PCR Experiment 1		Relative Expression PCR Experiment 2	
Gene	RNAseq	GAPDH	PGK1	GAPDH	PGK1
IL6	-	0.96 ± 0.07	0.88 ± 0.21	0.90 ± 0.15	0.84 ± 0.17
IL8	2.85	2.84 ± 0.04 ***	2.59 ± 0.25 **	2.86 ± 0.40 **	2.68 ± 0.59 **
JUN B	0.42	0.15 ± 0.11 **	0.14 ± 0.18 **	0.21 ± 0.02 **	0.19 ± 0.05 *

**Table 9 ijms-22-02084-t009:** Effect of NPs on the secretion of IL8 by T98G cells. T98G cells were exposed to 20 µg/mL of TiO_2_-NPs or to 5 µg/mL of ZnO-NP for 24 h or 72 h. Afterwards, IL8 determinations were made following ELISA procedures, as described in Section 4.13. Each experimental condition in each independent experiment was performed with three different wells. The IL contents in the control (unexposed) wells were considered 100%. The treatment done with 300 µg/mL of lipopolysaccharides from *Escherichia coli* O55:B5Y was taken as the positive control. The control cultures secreted after the 72-h exposure were 618-, 718- and 644-pg IL8/mL in experiments 1, 2 and 3, respectively. The control cultures secreted after the 24-h exposure were 282- and 370-pg IL8/mL in experiments 1 and 2, respectively. These experiments were run in parallel to the silver NPs and shared controls with the data published by Fuster and coworkers [15]. *** = statistically different from the control for *p* < 0.00; ** = statistically different from the control for *p* < 0.01 and * = statistically different from the control for *p* < 0.05.

	Sample	IL8 (%) in Experiment 1/2/3
72-h exposure		
	TiO_2_-NPs	128 **/133 **/143 ***
	ZnO-NPs	95/103/113 *
	Positive control	217 ***/223 ***/230 ***
24-h exposure		
	TiO_2_-NPs	146 */135 **
	ZnO-NPs	87 */101
	Positive control	315 ***/228

## Data Availability

The RNAseq data presented in this study are openly available in Sequence Read Archive (SRA) (https://www.ncbi.nlm.nih.gov/sra/, accessed on 29 December 2020) with Bio Project accession number PRJNA580150.

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
