# Peer review of "Titanium Dioxide, but Not Zinc Oxide, Nanoparticles Cause Severe Transcriptomic Alterations in T98G Human Glioblastoma Cells"

_ijms, 2021, doi:10.3390/ijms22042084_

Round 1
Reviewer 1 Report
Authors have done intensive studies of Titanium dioxide and zinc oxide on the effects of T98G human glioblastoma cells.
It is plausible authors provide plenty evidences to confirm the toxicity of Titanium dioxide and mechanism. Please arrange the manuscript following the suggestions.
The introduction.
Authors planed to do neurotoxicity, please introduce the link between neurotoxicity with your methods.
Authors have done lots of different methods please mention them in the introduction.
Results
Although you have done the TEM after Proteomic and gene expression, but you need to give readers a general idea of your results. The TEM results convince us and give a direct evidence. Can you please move the section 2.6.2. to section 2.2? The order should be first 2.6. 2 and then 2.6.1. You do not need mention that you did TEM at end of your experiments.
Method section
It is very surprised that the HPLC was firstly mentioned at line 719 of page 25.
4.12.2HPLC-MS/MS analysis
Minor errors
Table S5 and S6
Please include the number of samples in the table legend
Figure S3 and Figure S4
The scale bars of images are missing. Please include the scale bars.
Reviewer 2 Report
Fuster et al describe the effects of TiO2 and ZnO nanomaterials on T98G glioblastoma cells. After 72 hours of exposition, the transcriptome of the cells exposed to TiO2 but not ZnO revealed alterations in many molecular pathways. Further, there was an increase in interleukin release, suggesting a possible inflammation. The RNA seq and the proteomic results are interesting as both materials are currently used in different products. However, there are important issues, mostly related with the characterization that must be clarified in order for it to be accepted.
1- One of the most important issues is the aggregation of the nanomaterials. In fact, the initial size is close to 700nm, which is nearly in the micron range. Authors state that once in culture medium, there is a “size reduction phenomenom” and for instance ZnO nanoparticles are found at 6.5 and 12 nm. Can the authors explain how they can found a population at 6.5 nm when the mean particle size by TEM is 23.15 ±8 nm? Even when serum proteins can stabilize nanoparticles, the size can not be smaller than that reported by TEM. In fact, in water they observe populations of 1 nm. Could this be an artefact?.
It is known that measurements performed with DLS in medium containing FBS can be altered due to the protein content or the colour of the medium that can interfere with the laser wavelength. This, together with the aggregation and precipitation of the nanomaterials that the authors describe, may indicate that DLS measurements are not appropriate in this case. Thus, it is hard to conclude, “the effects herein reported can be attributed to materials on the nanometric scale”, that is <100nm. In the same line, it is difficult to conclude that these nanomaterials behave different to other nanomaterials with the same composition, as the aggregation can greatly affect the uptake and cytotoxicity. Authors should take this information into consideration and change the discussion accordingly.
2- According to MTT results authors chose 5uM of ZnO particles to perform the experiments. How many biological replicas were used when performing the MTT? From the materials and methods section it seems that only one replica with 6 wells was performed; there are measurements where only 4 wells were taken into account. From this assay the doses were chosen for the rest of the manuscript but differences of 3% in this assay can be not significant.
3- How were the particles sterilized before performing the experiments?
4- Again, it is important to note that if the particles aggregate in cell culture medium, it is difficult to compare between both compositions as it is not possible to know the percentage of particles that are interacting with the cells (without precipitating into the bottom of the well). I will suggest measuring the Zn o Ti content that remain in the supernatant after 24 and 72 hours of incubation. The precipitation or the low dose that was used can be related with the lack of accumulation of ZnO particles inside the cells. In fact, as the size and the z potential is similar, one will expect a similar amount of internalized particles. Even when nanoparticles can cause alterations without entering the cell (ion leaking for instance), in this case it is difficult to extract many conclusions. Probably, ZnO particles incubated at a higher dose would enter into the cell, although the mechanism could be different. Although it is not possible to perform this work now, it would be interesting to know what happen in that case. However, the discussion should include these considerations.
Round 2
Reviewer 2 Report
The manuscript can be accepted in its present form